

# Locating the Optimal Wind Resource within two Californian Offshore Wind Energy Areas

Arka Mitra [1], Virendra P. Ghate [1], and Raghavendra Krishnamurthy[2]

[1]Argonne National Laboratory, Lemont, IL, USA

[2]Pacific Northwest National Laboratory, Richland, WA, USA

*Correspondence to*: Arka Mitra (mitra@anl.gov)

**Abstract.** The spatiotemporal variability of wind resource at two Californian offshore wind-energy areas (Humboldt and Morro Bay) is characterized using 23 years (2000-2022) of 2-km-resolution, hourly NOW-23 reanalysis data. Idealized power is estimated for the International Energy Agency 15 MW
reference wind turbine. Wind speed and energy output are higher at Humboldt than Morro Bay and for summer months than winter months at both sites. Idealized daily energy output per turbine with one turbine within each model grid cell, peaks at 330 MWh in June for Humboldt and 300 MWh in May for Morro Bay. Energy output per turbine decreases from the oceanward to the coastward perimeter by ~20%, dropping ~22 MWh across Humboldt and ~46 MWh across Morro Bay. Rotor-layer wind shear and veer
exhibit strong seasonal variability, with summertime shear twice of wintertime shear at both sites. Daily wind resource variability is quantified through Fractional Variability (FV), defined as the ratio of the interquartile range of wind speeds/energy to the overall median value for that day of year. Locations and times with higher FV coincide with low wind-speeds (i.e., low output) for both sites. A linear optimization identifies the optimal wind resource locations (that maximizes energy output but minimizes output FV,
wind shear, wind veer, and distance to shore) at the oceanward and coastward flanks of Humboldt and Morro Bay, respectively. The gradients in optimization scores are aligned parallel to the coast and are independent of the choice of power curves for rated powers of 8-16 MW.

# 1 Introduction

The Californian coastline offers an expansive region for the proliferation of floating wind farms along the
US West Coast. Five wind energy areas (WEAs) spanning a total surface area of 1510 sq. km are being





considered for development off the coast of central and northern California (Musial et al., 2016; Beiter et al., 2020). Each WEA is projected to have 1,000-1,300 MW capacities (4.6 GW over all 5 sites). Given the vast sizes of these WEAs, it is expected that wind turbines situated at different sections of these wind farms would likely encounter spatiotemporal variability in meteorological conditions. Moreover, between

2025-2030, the target floating offshore wind turbine power capacities are expected to reach up to 25 MW, which are planned for hub-heights of 170 m and a rotor diameter of 280 m (Zahle et al., 2024). At this current rate of progress, wind turbine rotor layers may reach heights of above 300 m by the next decade. Such tall rotors will likely often interact with the tops of the planetary boundary layer (PBL), sometimes even the free troposphere and will likely often be submerged within cloud boundaries (for climatological

mean PBL heights in the region, see Burk and Thompson, 1996; Engeln and Teixeira, 2013). Thus, wind turbines within these offshore WEAs are likely to encounter time-dependent variability in wind conditions in both the horizontal and vertical dimensions. Since a climatological record of direct observations of rotor layer winds are not available from these sites, this study will be a first attempt to understand the measure of variability in wind resource within the proposed WEAs. Here, we specifically examine the

two proposed WEAs at Humboldt and Morro Bay, respectively.

There have been extensive research characterizing the wind resource off the general Californian coastline (e.g., Jiang et al., 2008; Lopez et al., 2022; Musial et al., 2016a; Wang et al., 2019, etc.). However, Mitra et al. (2024) was the first to focus primarily on the Humboldt and Morro Bay WEAs, by characterizing the hub-height winds in the region vis-à-vis the larger-scale climate (meteorology, clouds and

precipitation) variability of the Northeast Pacific (NEP) oceanic region. Analyzing long-term reanalysis and satellite data, that study found that the mean winds, meteorology, clouds, and precipitation at the Humboldt and Morro Bay sites are largely driven by the climate dynamics of the NEP high pressure system (Hickey, 1979). However, the study also noted that this mean state is strongly modulated by coastal phenomena such as cold-water upwelling and terrain-related phenomena which leads to strong

coastal gradients in surface fluxes, PBL heights and cloud cover and distinct annual cycles of temperature and moisture advections. Warm low clouds are persistent, with cloud fractions varying between 20-100% over a year, with average rain area fractions of 32% and 19% over Humboldt and Morro Bay respectively. Moreover, frequent instances were observed where the wind turbine rotor layers are either within satellite





estimates of cloud boundaries and/or above reanalysis estimates of the PBL tops, especially at Morro Bay
in summer months. These findings highlight the possibility of significant variability of wind-resource at
various points within the wind farms and even within individual wind turbine rotor layers.

Wind speeds and the electric power generated from wind farms can vary across different spatiotemporal
scales. Hence, in recent years, a lot of emphasis has been put on the ready availability of short-term (sub-
hourly to daily) wind speed forecasts at or near wind-turbine hub heights and wind turbine rotor layers
(Wilczak et al., 2019; James et al., 2022). As a result, such forecasts are now readily available from direct
numerical simulation and statistical post-processing. Variability within these forecasts on hourly and daily
timescales are largely driven by the inherent diurnal variability of the Earth's climate (occasionally
disrupted by random weather events, such as surface low-pressure systems). Hence, characterizing this
natural daily variability and the deviations from the same is crucial in designing the layout of wind
turbines, estimating the electricity produced by wind farms over their lifetime and in devising strategies
to improve the lifetime of the farms in question. As a first step to characterize this within-WEA variability
of wind resource and understand its meteorological controls, we shall leverage model reanalysis data at a
higher spatial resolution than was done in Mitra et al. (2024). Existing mesoscale wake parameterizations
(Fitch et al., 2012) mis-represent internal-wake effects and do not accurately capture cluster-wake effects
downstream of a wind farm (Gomez et al., 2024). The impact of wakes from offshore floating platforms
is a point of active research (Angelou et al., 2023). Therefore, assuming 'idealized' power (i.e., no wind-
turbine wakes and other expected losses), we shall investigate the spatiotemporal variability of wind
resource within the Humboldt and Morro Bay WEAs. This analysis is done with the explicit goal of
searching the locations within the WEAs with the most and least optimal wind resource. Such a study is
a crucial first step towards future analyses that will investigate the preferred siting and layout of wind
turbines within these WEAs. Moreover, such a study will also inform future studies involving numerical
weather models simulating winds and associated conditions at these sites.

The details of the dataset used, the concepts of 'idealized' power and the metric we use to calculate
variability of wind resource and how we define the location of optimal wind resource follow in Section





2. Estimates of spatiotemporal variabilities in wind resource (idealized wind power, wind speeds etc.) and the location of the optimal resource is given in Section 3. Concluding discussions follow in Section 4.

# 2 Data & Methods

## 2.1 Data

The two Californian offshore WEAs studied here are located near Humboldt and Morro Bay, with their central coordinates at ~41°N, -125°E and ~36°N, -122°E, respectively. The northern WEA near Humboldt covers an area of 536 km$^2$ and its nearest and farthest distances from the coast are ~35 to 55 km. The water depths at Humboldt vary between 500-1100 m, with steeply sloping seafloor bathymetry (gradients $\geq 4°$) towards the western boundary. Meanwhile, the southern WEA near Morro Bay covers an area of 975 km$^2$ and its nearest and farthest distances from the coast are ~32 to 60 km. Morro Bay have deeper water depths (800-1300 m), but has a gentler sloping seabed than Humboldt (Cooperman et al., 2022).

The data used to characterize the wind resource and variability of these WEAs is the NOW-23 reanalysis (Bodini et al., 2024). This is the latest version of the Wind Integration National Dataset (WIND) Toolkit (Draxl et al., 2015), a dataset often leveraged by the offshore wind industry for wind profiles around the coastal US at high spatiotemporal resolution. This current version of the product (NOW-23, hereafter) implements the Weather Research Forecasting (WRF) model to simulate wind profiles up to 500 m above surface level for coastal regions from January 1, 2000, onwards. NOW-23 significantly differs from earlier iterations of WIND Toolkit as the WRF model here is forced with ERA5 Reanalysis atmospheric profiles (Hersbach et al., 2020) and the OSTIA sea-surface dataset (Donlon et al., 2012). For the North Pacific region (containing the Californian offshore sites), NOW-23 data currently extends through December 31, 2022. For this study, we downloaded the 2 km horizontal spatial resolution and 1-hour time resolution NOW-23 files (1 file for each year between 2000-2022) available through the Amazon Web Service (AWS) public data registry page of the U.S. Department of Energy. The 2 km-resolution NOW-23 is resampled to a 0.02-degree equal area grid, on which further analyses are done.



Wind power output is estimated using the IEA 15 MW offshore reference wind turbine (RWT;
Supplementary Figure S1) power curve (Gaertner et al., 2020), assuming a solitary turbine at each grid
point. Thus, we consider 135 independent wind turbines at Humboldt and 204 independent wind turbines
at the Morro Bay WEA. Turbines reach maximum power (15 MW) between wind speeds of 10.3-25 m
s$^{-1}$, with cut-in and cut-out speeds of 3 m s$^{-1}$ and 25 m s$^{-1}$, respectively. We only consider the steady-state
power curve, ignoring the effects of wind-turbine wakes, as this exploratory study focuses on identifying
the optimal unperturbed wind resource. Daily aggregate energy output is derived by summing hourly
energy output (in MWh from power output) for each grid cell. The total energy output of the wind farms
(Humboldt or Morro Bay) is calculated by summing the energy output of all grid cells within the WEAs.

For each grid cell, we defined wind shear and veer as the absolute changes in wind speed (m s$^{-1}$) and
direction (°) over the RWT rotor layer (30 m to 270 m A.S.L., with hub-height at 150 m A.S.L.), with
negative wind veer indicating a counterclockwise rotation. Typically, the wind-energy industry adopts
estimates of shear and veer that are normalized by heights of rotor edges (e.g., bulk rotor-layer veer, $\beta = \frac{\theta_2 - \theta_1}{z_2 - z_1}$, where $\theta_2$ and $\theta_1$ are the wind directions at heights $z_2$ and $z_1$, respectively). However, for this study
we simply choose to study the raw differences at the top and bottom of the turbine rotor layer. This is
done as we consider only one height interval (i.e., the rotor-layer depth) and raw values are more intuitive.

## 2.2 Optimal Wind Resource

The objective of this study is to find the locations of optimal wind resource within the Humboldt and
Morro Bay WEAs. Our primary consideration is the direct influence of meteorological conditions,
specifically the variability in turbine hub-height wind speeds. Locations with consistently high average
wind speeds typically yield more reliable power. However, minimizing power variability is also critical
to ensure stability and reduce grid integration costs. Fluctuations in wind speed complicate power output,
requiring expensive and non-renewable backup or energy storage systems (such as battery or pumped
hydro storage). Additionally, deviations from simple hub-height wind speed-to-power output
relationships arise due to variations in wind vectors over the turbine's rotor layer. Studying wind shear
and veer variability across the WEAs allows us to assess the limitations of our idealized power estimates.





130  Economic considerations, such as capital and operational costs, are also important to consider but are typically secondary during early site development. For this study, we use distance from the shore as a proxy for all such costs (Ren et al., 2021). Thus, after all these considerations, we define the location of optimal wind resource for a WEA as the grid cell that simultaneously maximizes power output while minimizing power variability, wind shear and veer across the rotor layer, and distance from the shore.

## 135  2.2 Variability Metric for Wind Resource

Several methodologies have been used to characterize the spatiotemporal variability of wind resource within wind farms, each with varying strengths. Lee et al. (2018) compared 27 such metrics and found that many are sensitive to outliers, especially with non-Gaussian wind data prone to extreme events. Among these, the ratio of the interquartile range (IQR) to the median was found to be the most robust and

140  resistant to outliers, maintaining statistical resilience over time. Given the hourly resolution of our dataset, we adopt a similar metric to study the diurnal variability of wind resource and its deviations.

We define daily Absolute Variability (AV) at each grid cell as the IQR (between the 95th and 5th percentiles) of the hourly wind resource for that day. This AV is normalized by the median wind resource for that day of year (DOY) over the wind farm, giving the Fractional Variability (FV):

$$FV(day, grid\ cell) = \frac{AV\ (day, grid\ cell)}{Median\ Value\ for\ all\ grid\ cells\ for\ that\ DOY}$$

145

This normalization accounts for daily and seasonal variations, with the median providing a robust central tendency. FV is calculated for each grid cell and day in our dataset to quantify spatiotemporal variability. Interpreting FV is straightforward: values of FV >> 1 indicate high variability relative to the median, suggesting greater susceptibility to model uncertainty in wind resource predictions.

## 150  3 Results

The analysis is divided into three subsections – in sections 3.1 and 3.2 we discuss the climatology of daily aggregate wind energy at Humboldt and Morro Bay lease areas and the spatiotemporal variability of that



idealized energy output. In Section 3.3, a simple linear optimization technique is used to ascertain the locations of the most optimal wind resource within these WEAs.

## 3.1 Climatology of Daily Aggregate Wind Energy

The annually averaged daily energy output per grid cell is greater at the Humboldt (206 MWh) WEA than the Morro Bay WEA (182 MWh). Although the annually averaged hub-height wind speeds are northwesterly at both sites (Fig. 1), the WEA-wide averaged hub-height wind speeds are slightly stronger at Humboldt than Morro Bay (12.3 and 8.7 m s$^{-1}$, respectively), leading to the greater energy output at the Humboldt WEA compared to Morro Bay WEA.

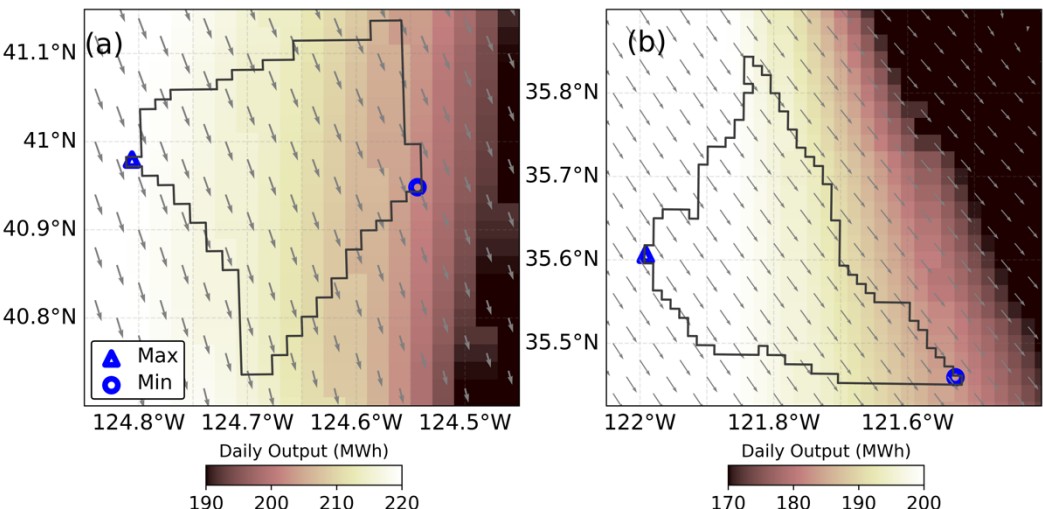

**Figure 1.** Annually averaged daily energy output (colors), hub-height wind speeds (arrows), and the locations of the grid-cells with maximum (MAX) and minimum (MIN) overall energy output for the (a) Humboldt and (b) Morro Bay WEAs.

Seasonal wind speeds and energy output at Humboldt and Morro Bay exhibit distinct spatial variability, with higher values in summer (June-July-August or JJA) than in winter (December-January-February or DJF) (Supplementary Figure S2). Wind patterns are closely tied to the seasonal dynamics of the Northeast Pacific (NEP) High Pressure System (Checkley and Barth, 2009). In Humboldt Bay, winter (DJF) winds show ~70% easterly flow resulting in a mean southeasterly flow which transitions to a predominantly





northwesterly flow (>40%) in summer (Supplementary Figure S3). Morro Bay, located further south, experiences consistent northwesterly winds year-round (Supplementary Figure S4).

Energy output at both WEAs (both in annual and seasonal means; Fig. 1 and Supplementary Fig. S2) is highest over open oceans and lowest closer to the coast, reflecting the steep descent in PBL heights that is characteristic of this coastline (e.g., see Figure 3 of Jiang et al., (2008)). This descent is caused by the

combined influence of the NEP high-pressure system over the ocean and the continental low-pressure system over land but is also modulated by coastal influences like cold-water upwelling and terrain. Wind energy output gradients are aligned parallel to the coastline at both sites (Fig. 1). The range of average energy output per grid cell is greater at Morro Bay due to its bigger size than Humboldt. Another reason may be its relative proximity to the coast (grid cells at Morro Bay are ~10 km closer to shore on average

as compared to Humboldt). These gradients of wind energy output per grid cell for are aligned parallel to the orientation of the coastline irrespective of season for both WEAs (north to south for Humboldt and northwest to southeast for Morro Bay, respectively). The drop in energy output per grid cell from the west to east perimeter at Humboldt Bay is 22 MWh, compared to a more pronounced 46 MWh drop from southwest to northeast perimeters at Morro Bay, indicating greater spatial variability at Morro Bay.

Grid cells with high frequency of occurrence of maximum (MAX) daily aggregate energy for a given WEA are typically located oceanward, while grid cells with minimum (MIN) daily aggregate energy are typically situated coastward (Figure 2). The average frequency of occurrence of MAX output for grid cells on the westward perimeter of Humboldt Bay is 12%, whereas for Morro Bay it is 7%. The average frequency of occurrence of MIN output for grid cells on the eastward perimeter of Humboldt Bay is 16%,

whereas for Morro Bay it is 11%. These locations of MAX and MIN outputs are not static and vary with shifts from the dominant wind directions (Supplementary Figures S3 and S4). Hence, there are also instances where it is possible to find MAX output on the eastward perimeter and MIN output on the westward perimeter for both WEAs. Such deviations are more common at Humboldt Bay, where weather events (e.g., passing fronts or atmospheric rivers) can disrupt surface pressure gradients, leading to more

cyclonic wind patterns. These weather disruptions, more common during winter (Supplementary Fig. S3a), reduce the clear separation of MAX and MIN points. As a result, there is a 1-5% probability of




finding MAX or MIN output in interior grid cells for Humboldt. In contrast, Morro Bay shows less variability, with <1% probability of MAX or MIN outputs occurring for interior grid cells.

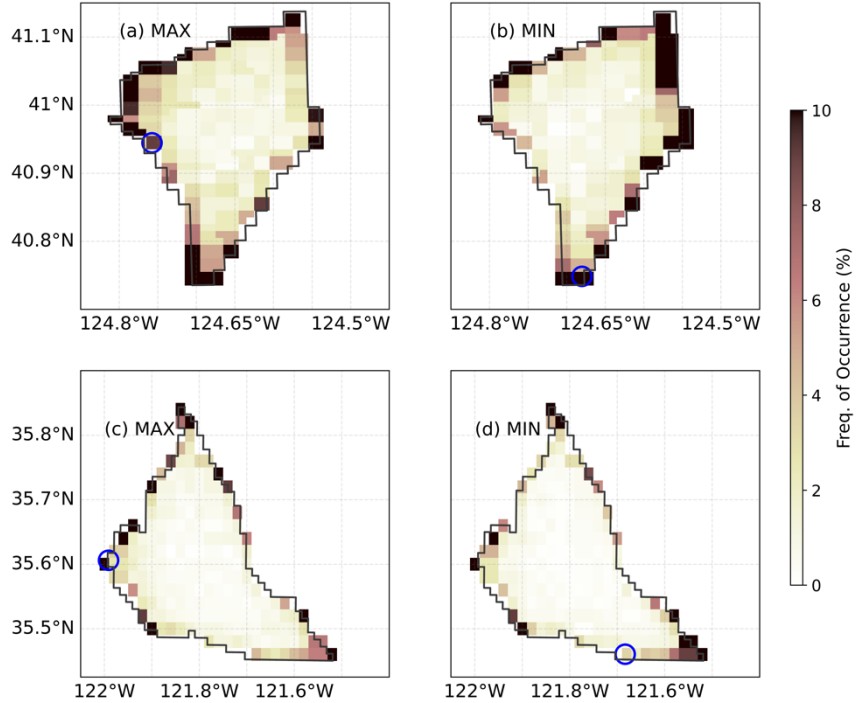

**Figure 2.** Frequency of occurrence of (a) MAX and (b) MIN daily aggregate energy output at Humboldt Bay and frequency of occurrence of (c) MAX and (d) MIN daily aggregate energy output at Morro Bay wind energy lease area. For each panel, the grid-cells with the most occurrences of HOURLY maximum and minimum energy output are marked.

The locations of the greatest frequencies of occurrences of MAX and MIN daily aggregate energy outputs do not necessarily coincide with the locations of overall maximum and minimum energy production. Instead, the grid cells with maximum and minimum net aggregate energy output at both WEAs are located at the westward and southern edges of the WEAs, respectively (Fig. 2). Supplementary Figures S5 and S6 show the time variation of daily aggregate energy output for the last 2 years in our dataset for these MAX/MIN pairs of grid cells for the two WEAs. Over these two years, while the power output for MAX and MIN points within each lease area fluctuates in phase, their magnitudes can vary substantially (especially in winter months). The difference in mean daily aggregate energy between MAX and MIN





points is 18% for Humboldt Bay (205 MWh vs. 242 MWh; Supplementary Figure S5) and 22% for Morro Bay (204 MWh vs. 167 MWh; Supplementary Figure S6).

The spatial patterns of frequencies in Fig. 2 largely follows the spatial variations in frequency of occurrence of hours with zero power output versus hours with rated power output (i.e., 15 MW) for a given grid cell (Figure 3). There is low spatial variability of No Power output at Humboldt (average = 8.3%) and in Rated power output at Morro Bay (average = 39.8%). However, there is a marked increase from ocean to coast in the frequency of occurrence of No Power at Morro Bay (average = 14.5%) and Rated Power at Humboldt (average = 46%).

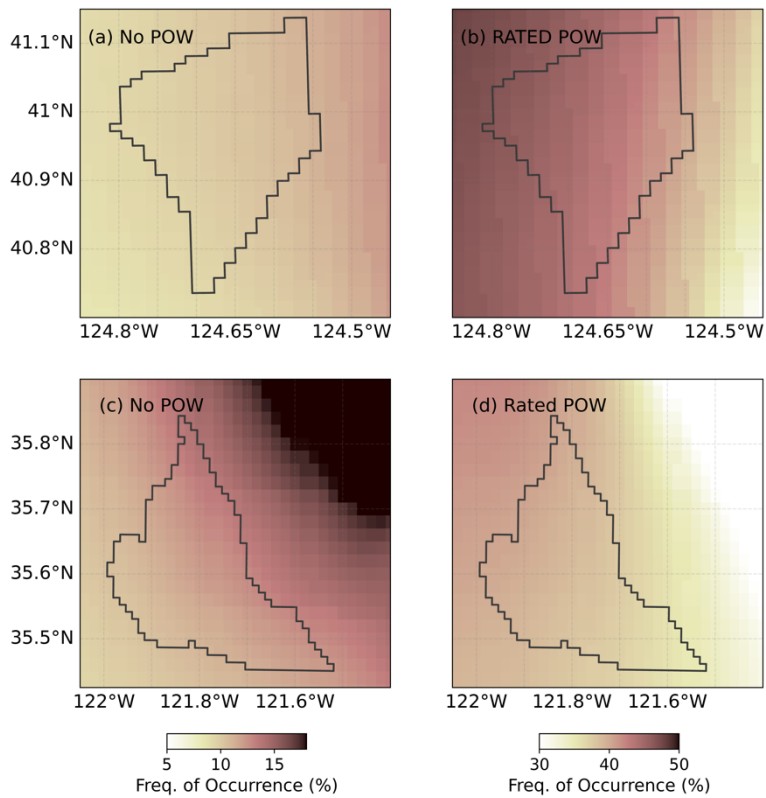


**Figure 3.** Frequency of hourly occurrence of (a) No power and (b) Rated power at the Humboldt WEA and of (c) No Power and (d) Rated power power at the Morro Bay WEA.

Our estimates of wind energy output have been derived assuming that the hub-height wind speed is sufficient to estimate the wind turbine power. However, both the speed and direction of wind can change





with height across the rotor layer swept by the turbine blades, potentially leading to deviations from our power estimates (Choukulkar et al., 2016). Mean $\pm$ Standard deviation of rotor-layer wind shear for Humboldt and Morro Bay are 2.1 $\pm$ 0.1 and 0.7 $\pm$ 0.3 m s$^{-1}$, respectively. Meanwhile, mean $\pm$ Standard deviation of rotor-layer wind veer for Humboldt and Morro Bay are -6 $\pm$ 20 and -7 $\pm$ 22°, respectively. Annually averaged wind shear and veer increases from the northwest to southeast at Humboldt, whereas

they increase northeast to southwest at Morro Bay (Fig.4). On average, winds are stronger at the top of the rotor layer than at the bottom, with an overall counterclockwise rotation for both wind energy areas.

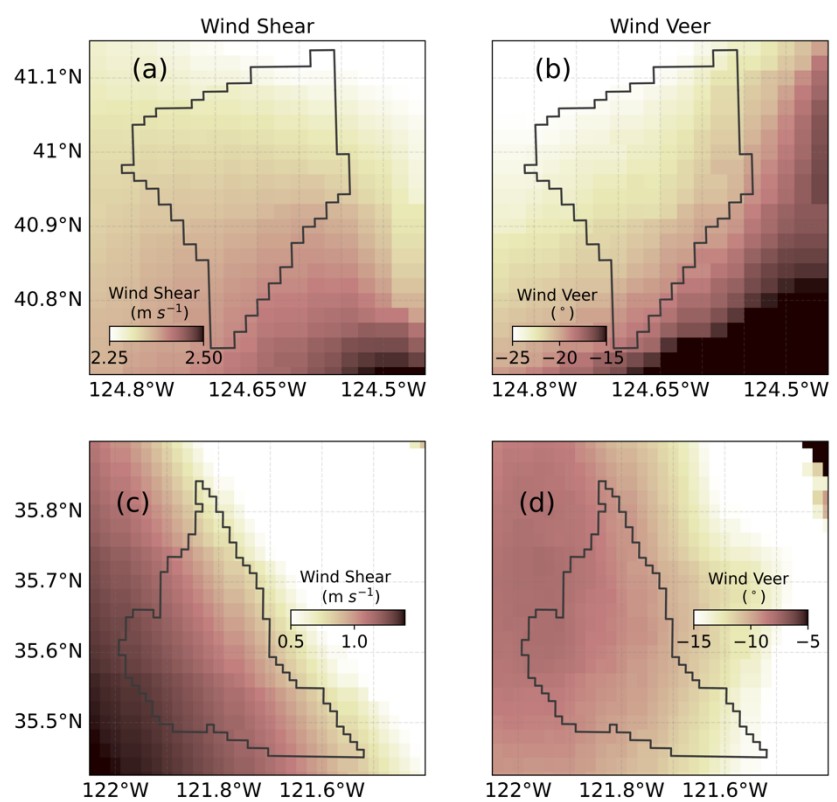

**Figure 4.** Annually averaged rotor-layer (a) wind shear and (b) wind veer for the Humboldt WEA and rotor-layer (c) wind shear and (d) wind veer for the Morro Bay WEA.

Seasonal mean wind shear at both WEAs exhibits distinct variability, with summertime shear (3.9 $\pm$ 0.4 m s$^{-1}$ for Humboldt and 1.6 $\pm$ 0.3 m s$^{-1}$ for Morro Bay) being nearly twice as much as wintertime shear (1.7 $\pm$ 0.1 m s$^{-1}$ for Humboldt and 0.8 $\pm$ 0.2 m s$^{-1}$ for Morro Bay) (Supplementary Fig. S7). The





occurrences of high versus low shear conditions across the Humboldt and Morro Bay WEAs are tied to the comparatively lower versus higher frequencies of occurrence of at least half of the wind turbine rotor layers in the free troposphere (Supplementary Figure S8). For both WEAs, this probability varies from ~30% to ~40% from the oceanward to the coastward perimeter of the wind-lease areas, following the steady descent of the well-mixed PBL. Seasonal mean wind veer at Humboldt and Morro Bay also shows distinct patterns, but unlike shear, the seasonal contrasts are not similar for both sites (Supplementary Fig. S9). Humboldt experiences an identical backing profile (counterclockwise rotation) across the rotor layer throughout the year (Fig. S9 and S9b). In contrast, Morro Bay experiences stronger backing during winter, with veer reaching -25° in the southeastern edge of the WEA (Fig. S9c), but much weaker veer between 0° and -5° in summer (Fig. S9d). Sharp gradients in shear and veer are present in summer at Humboldt and winter at Morro Bay, particularly closer to coast. This suggests shifts in wind vectors over different sections of the rotor layer for different grid cells.

The co-variability of these absolute values of shear and veer with more industry-specific terms are provided in Supplementary Figures S10 and S11. The absolute value of log-law shear, $\alpha$ exceeds 0.5 less than 10% of all time at both WEAs and the absolute value of bulk veer, $\beta$ exceeds 0.25 less than 4% of the time at both WEAs. While infrequent, these occurrences highlight scenarios where hub-height winds may not be sufficient to estimate wind energy output. However, the deviations of "true" output (such as derived using a rotor-equivalent wind speed) from our estimates are beyond the goals of this current work. A simple analysis shows that considering the combined effect of wind shear and veer can lead to deviations from our estimates of idealized output (especially through the impact of shear) (Supplementary Figure S12). However, accurately resolving these effects would require sub-hourly-resolution data to estimate the short-term deviations of wind speed and direction across the rotor layer, which are left to future work. For this study, we shall treat shear and veer simply as sources of uncertainties in energy output estimates, and hence as variables to be minimized in our optimization (Section 3.3).

Over the entire dataset, the median hub-height wind speed and daily aggregate energy output at Humboldt Bay (9.4 m s$^{-1}$ and 225.6 MWh, respectively) are typically higher than at Morro Bay (7.2 m s$^{-1}$ and 192 MWh, respectively). Monthly median wind speeds (Fig. 5a) vary between 6 to 11 m s$^{-1}$ at Humboldt and





5 to 10 m s$^{-1}$ at Morro Bay, peaking in June at both sites. Despite a mild annual cycle in wind speeds, the daily aggregate energy output per grid-cell is very sensitive to small variations in wind speeds (Fig. 5b). The portion of the RWT power curve that is most sensitive to wind speed changes (i.e., between 3 to 10.2 m s$^{-1}$) witnesses an average 1.8 MW power output change for every 1 m s$^{-1}$ change in hub-height wind speeds (Supplementary Figure S13). Hence, there is a pronounced annual cycle in energy output, with

peak daily output of ~330 MWh per grid cell (~44.5 GWh over the entire area) at Humboldt in June and July, and ~300 MWh per grid cell (~61 GW overall) at Morro Bay in May and June. There are also instances of no daily aggregate power for all months of the year at both sites since hub-height wind speeds less than the cut-off wind speed for the IEA 15 MW RWT can occur at both sites throughout the year.

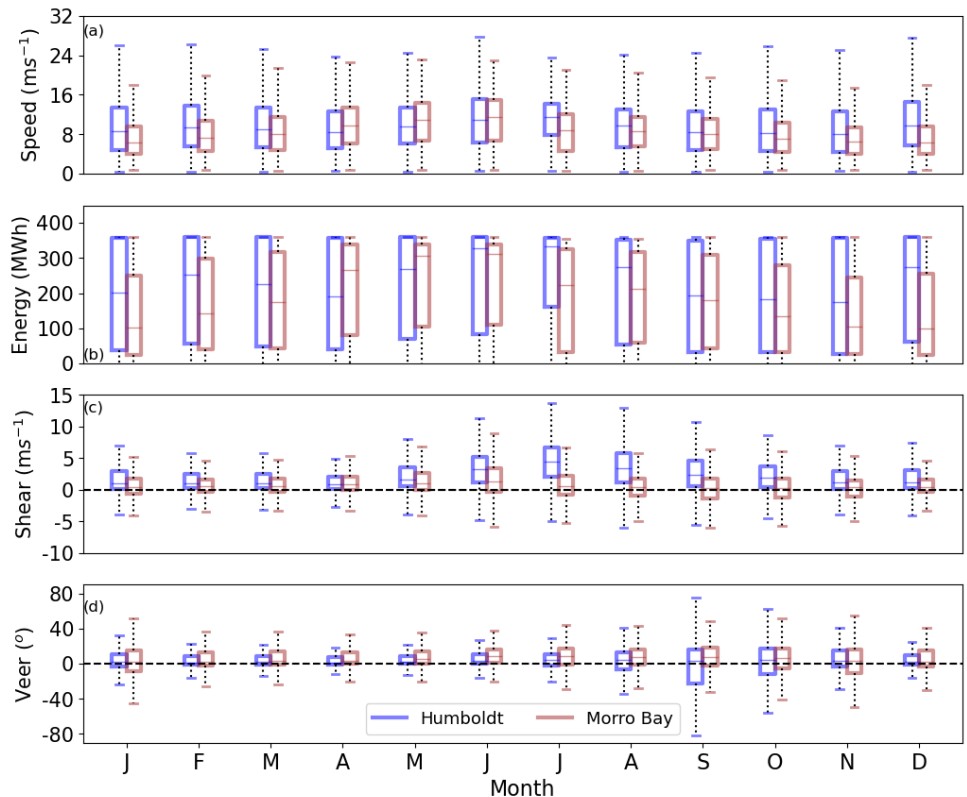

**Figure 5.** Monthly variation of (a) Hub-height (150 m A.S.L) wind speeds, (b) Power (MW), (c) Planetary Boundary Heights (km), (d) Rotor-Layer Wind Shear (m s$^{-1}$) and (e) Rotor-Layer Wind Veer (°) at Humboldt (blue) and Morro Bay (brown) wind energy lease areas. Shown values are the daily estimates from all grid cells within the respective WEAs.



Monthly median rotor-layer wind shear (Fig. 5c) shows a sharp annual cycle, with Humboldt Bay
experiencing higher shear than Morro Bay throughout the year. Shear peaks at Humboldt Bay (~5 m s$^{-1}$)
between May-August, while at Morro Bay it remains below ~2 m s$^{-1}$. This is most likely tied to a stronger
coastal low-level jet at Humboldt than Morro Bay during summer (Sheridan et al., 2024). Monthly median
rotor-layer wind veer (Fig. 5d) presents a relatively consistent pattern across both sites, with near-zero
values for most months. However, equally frequent occurrences of backing and veering veer are observed
at both sites (25$^{th}$ and 75$^{th}$ percentiles up to $\pm$15°) are noticed in fall and winter (September-February).
Although wind energy deviations due to high shear/veer is left for future work, the spatiotemporal
variability in wind resource at the two WEAs due to the underlying meteorology is studied in Section 3.2.

## 3.2 Spatiotemporal variability of Daily Aggregate Energy

Figure 6 shows that the seasonally averaged Fractional Variability (FV) of daily energy output at both
Humboldt Bay and Morro Bay WEAs also follow similar ocean-to-coast gradients observed in daily
aggregate power (Fig. 1). However, FV increases ocean-to-coast instead of coast-to-ocean for energy.
Hence, the oceanward grid cells experience less wind variability and yield higher power as compared to
those near the coast.

At higher hub-height wind speeds (between 10.3-25 m s$^{-1}$), the RWT power curve is saturated at its full
capacity of 15 MW (Supplementary Figure S13). In this regime, there is no change in power output with
changes in wind speed. Hence, it follows that grid cells and times which experience such high wind speeds
also exhibit low overall variability in wind resource. It should be noted here, then, that our results are
therefore entirely dependent on two metrics – the wind speed distribution at the location and our specific
choice of reference wind turbine.



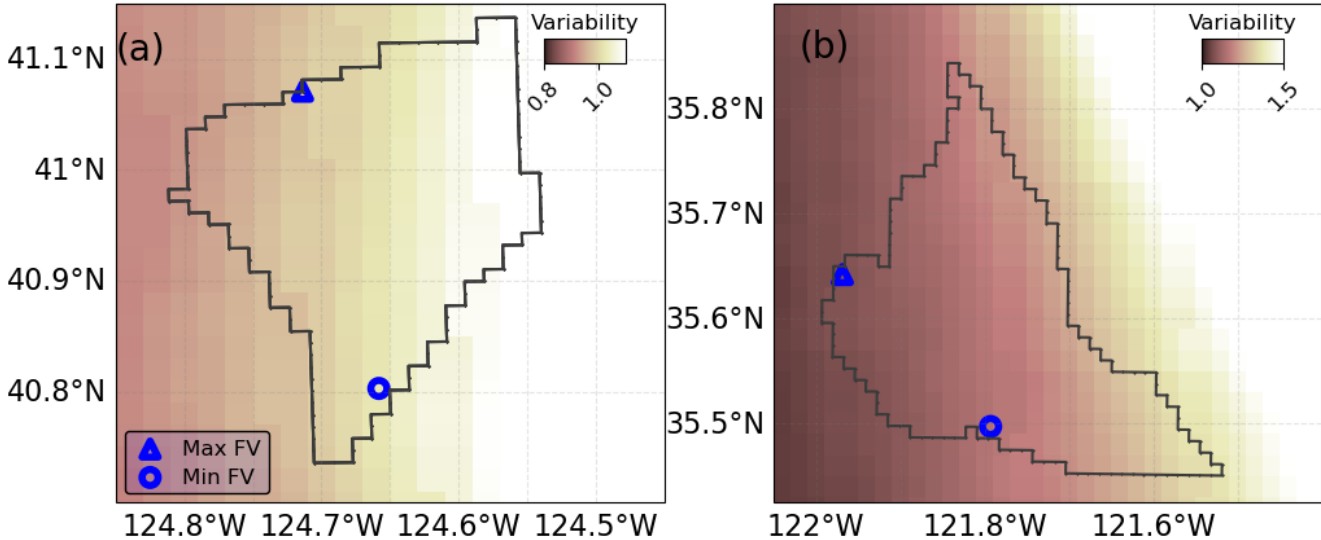

**Figure 6.** Annually averaged fractional variability (FV) of daily aggregate energy output (expressed as multiples of median energy output), and the locations of the grid-cells with maximum (MAX) and minimum (MIN) overall power output FV for the (a) Humboldt and (b) Morro Bay WEAs.

Power output and wind speeds are higher in summer (JJA) due to low variability (Supplementary Figure S14). The oceanward grid cells with higher daily power exhibit lower variability. In winter months (Supplementary Fig. S14a and S14c), FV > 1 occurs in several grid cells of both lease areas, implying large deviations in daily variability due to diurnal cycle that is far greater than the daily average. For Humboldt Bay, the highest FV values (>1.3) are concentrated in the southwestern regions, while oceanward locations show FV < 1. Similarly, Morro Bay's highest FV (>1.8) occurs in the southeast during winter. In summer (Supplementary Fig. S14b, S14d), FV values drop below 1 in most oceanward cells, indicating stable wind conditions at sub-daily timescales. The locations which experience highest variability in wind energy output are located in the perimeter of the WEA that is closest to the coast (east for Humboldt and southeast for Morro Bay). Meanwhile, the locations within the WEAs that experience minimum variability are consistently located more oceanward, further indicating that the variability of wind resource (speed and energy output) decreases from coast to ocean.





These patterns are also apparent in the monthly cycles of FV of wind speeds and energy output (Figure 7). FV is consistently higher in winter in both WEAs. For wind speeds (Fig. 7a), FV reaches 1.2 in January and drops below 1 (~0.6) during June-July at both sites. The variability in energy output (Fig. 7b) mirrors the variability in wind speed patterns, peaking at 1.3 for Morro Bay and 1.1 for Humboldt in winter, then dropping below 0.8 in summer for both WEA. Supplementary Figure S15 shows that the median monthly Absolute Variability (AV) of hub-height wind speeds ranges from 2 to 6 m s$^{-1}$, with the highest values in January at Morro Bay. Median AV of wind energy output ranges from ~100 MWh in summer to ~250 MWh in winter, with the variability in Humboldt being marginally higher. Thus, we notice a clear seasonality of FV and AV for both WEAs with winter months exhibiting higher variability, particularly at Morro Bay. Higher variability in wind energy outputs are typically associated with lower wind speeds and lower shear across the wind turbine rotor layers (Supplementary Figure S16).

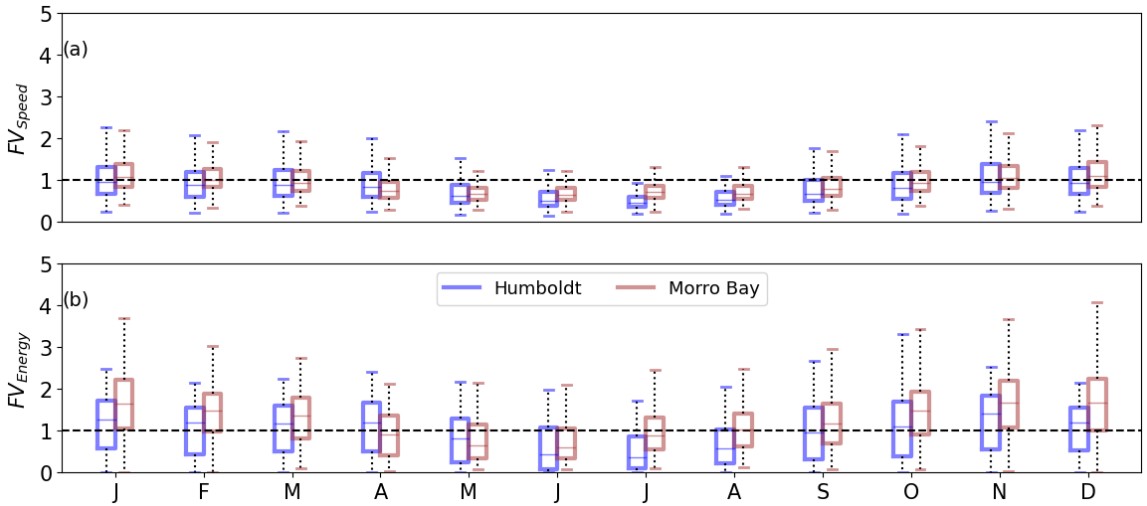

**Figure 7.** Monthly variation of fractional variability (FV) of (a) hub-height wind speeds and (b) daily energy output at Humboldt (blue) and Morro Bay (brown) wind lease areas.





# 3.3 Location of Optimal Wind Resource

To identify the locations of optimal wind resource within the Humboldt Bay and Morro Bay wind energy areas, we employed a gradient descent optimization algorithm to solve a multi-objective linear optimization problem. The objectives are:

1. Maximizing wind energy output,
2. Minimizing wind energy variability,
3. Minimizing wind shear and veer across the rotor layer, and,
4. Minimizing distance to shore (proxy for all costs).

The optimization score for each grid cell was calculated using the following equation:

$$S = w_{Out}.Out - w_{FV}.FV - w_{shear}.shear - w_{veer}.veer - w_{dist}.dist$$

where $S$ is the raw score for each grid cell (alternatively can be interpreted as a cost function), $Out$ is the normalized mean daily energy output, $FV$ is the normalized mean fractional variability of daily aggregate energy output, and $dist$ is the normalized distance from the corresponding coastal port, $shear$ and $veer$ are the normalized rotor-layer wind shear and veer, respectively. The weights ($w_{out}, w_{FV}, w_{shear}, w_{veer}$ and $w_{dist}$) control the relative importance of each term in the equation. Initial weights were chosen to reflect practical considerations in wind farm development. We choose $w_{out} = w_{FV} = w_{veer} = w_{shear} = 1$ to ensure that equal importance was given to maximizing energy output while simultaneously minimizing its variability (either direct or through deviations caused by shear/veer). Meanwhile, we chose $w_{dist} = 1.5$ to penalize the grid-cells farther from the coastline as they would incur greater costs for building, maintenance and repairs, irrespective of the meteorological conditions existing there.

The algorithm iteratively refined the score over 100 iterations with a learning rate of 0.01. Gradients were computed for each grid cell based on the partial derivatives of the score with respect to output energy, variability, distance, shear, and veer, adjusting the values to minimize the score. The final raw scores were normalized to a 0-100 scale, with higher scores representing grid cells with the most favorable conditions for turbine placement. The optimization results are shown in Figure 8.

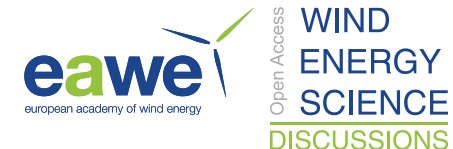

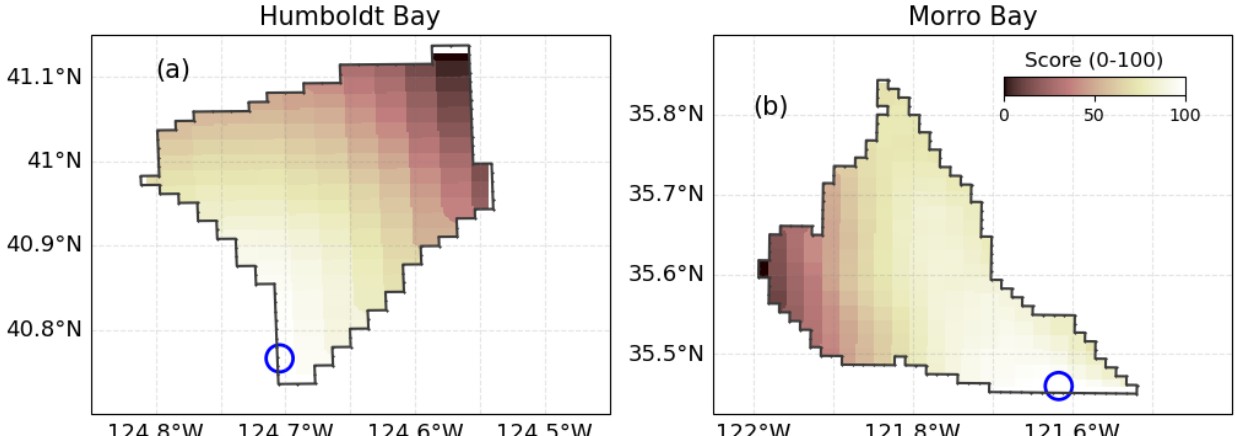

**Figure 8.** Linear Optimization Score (0-100) for grid cells at (a) Humboldt and (b) Morro Bay (brown) wind energy lease areas. The locations of the most optimal wind resource (Score = 100) at each wind energy lease area are marked with a blue circle.

At Humboldt Bay (Fig. 8a), the highest-scoring grid cell is located at approximately (40.75°N, 124.70°W) in the southwestern portion of the lease area. The scores decrease as we move coastward from the southwestern part of the WEA (associated with higher energy output and low variability) to the northeastern part of the WEA (lower energy output but higher variability). For Morro Bay (Fig. 8b), the most optimal grid cell is found at approximately (35.46°N, 121.62°W), in the southeastern part of the lease area. Unlike Humboldt Bay, the scores decrease while moving oceanward with the highest scores in the southeastern and the lowest scores in the western edges of the WEA. The difference between the alignment of gradients in scores between Humboldt and Morro Bay may be since the direction of gradients of wind shear and veer at the Morro Bay WEA is aligned opposite to the direction in which the gradients of wind energy output are aligned (comparing Fig 1b with 4c and 4d). The other important factor likely driving the directionality of the scores of the two WEAs are the shapes of the two areas, which varies the distances to the coast for specific grid-cells. To ensure that the scores shown in Figure 8 are not merely a by-product of our specific choice in wind turbine, a simple sensitivity analysis was conducted.

By randomly varying the 4 important metrics of a wind turbine power curve (i.e., the cut-in and cut-out wind speeds and rated wind-speeds and power), we generated 25 reference wind turbines with rated powers between 8-16 MW (Supplementary Figure S17). This choice is partly unrealistic because we did





not ensure that such wind turbines could mechanically accommodate the same hub-height and rotor layer dimensions as our IEA 15 MW RWT. However, since our goal with this exercise was to investigate the sensitivity of Figure 8 scores to choice of power curve, these random perturbations of the wind turbine

should suffice. As evident from Supplementary Figure S18, the different choices of wind turbines do not perturb the locations of optimal wind resource significantly. In fact, while the east-west separation of optimal-non-optimal grid cells in Morro Bay get slightly less prominent, the separation of points in the southwest-northeast direction for Humboldt does not see much discernible changes. Thus, for idealized wind energy with wind turbines between rated powers of 8-16 MW, our optimization scores are

independent of the choice of power curve (and wind turbines, by extension).

Thus, by focusing on high-scoring regions from our optimization technique, one can identify areas most favorable for wind turbine placement. However, additional optimization must also supplement this process – for example, accounting for extreme events, wind-turbine wakes, ocean floor bathymetry, waves and swells, combinations of various wind turbine infrastructures and specific wind turbine layouts could

potentially also give rise to significant deviations from the findings in this section.

# 4 Summary & Discussion

In this study, we characterized the spatiotemporal variability of wind resources at the Humboldt Bay and Morro Bay wind energy areas using high-resolution (2-km, hourly) NOW-23 reanalysis data (2000-2022). By employing an idealized power curve from the IEA 15 MW Reference Wind Turbine (RWT), we

quantified the climatology of wind energy output and its spatiotemporal variability. We used these findings to further locate the optimal wind resource at both WEAs, based on maximizing energy output while minimizing power variability, wind shear/veer and distance from shore (proxy for economic costs). The main findings of this study are:

a)    Wind speeds and energy output peak in summer (JJA) at both WEAs. Daily energy output per turbine

peaks at 330 MWh in June for Humboldt and 300 MWh in May for Morro Bay.

 

b) Energy output decreases oceanward to coastward within the WEAs, driven by gradients in wind speeds and PBL heights. Between the oceanward and coastward perimeters, there is a ~22 MWh drop in energy output per wind turbine in Humboldt and a ~46 MWh drop in Morro Bay.

c) The difference in mean daily energy outputs between the turbines with maximum and minimum
overall energy output is 18% at Humboldt Bay (205 MWh vs. 242 MWh) and 22% at Morro Bay (204 MWh vs. 167 MWh). This difference is more pronounced in winter due to weather disturbances, which can lead to sharp changes in the dominant wind directionality.

d) Fractional variability (defined as the ratio of daily interquartile range in wind resource to day-of-year median wind resource) is greatest in winter, peaking at ~1.4 for wind speeds and ~1.3 for daily energy
output. Spatially and temporally, higher variability coincides with weaker winds and lower outputs.

e) Seasonal mean rotor-layer wind shear shows distinct patterns, with summertime shear nearly twice that of winter for both sites. Veer patterns differ by site, with consistent backing at Humboldt and stronger winter backing at Morro Bay.

f) Simple analysis suggests the optimal wind resource is located at the southwestern edge of the Humbolt
WEA and at the southeastern edge of the Morro Bay WEA. The location and orientation of the optimization scores were found to be largely invariant to choice of wind turbine power curve.

Our findings provide valuable insights for both scientific research and practical applications in the wind farm development of these Californian wind lease areas. These findings have important implications for future modeling efforts, observational studies, and turbine siting strategies. We have also presented a new
metric - Fractional Variability (FV) - to capture wind resource variability. Although, we only analyzed FV of these two specific Californian offshore sites only, this approach can easily be employed in other wind farms around the world to quantify wind resource variability.

Although some locations within our WEAs were found to experience significant gradients and high absolute values of wind-veer and shear for specific seasons, the average values of both quantities over the
entire WEAs were found to be small (Supplementary Figures S10 and S11). The impact of wind-veer and shear on wind power output is likely to be strongly tied to rotor-layer wind vectors and are also likely co-variant (Murphy et al., 2020; Gao et al., 2021). For example, high shear may not automatically overlap



with orthogonal flow to the turbines and may not necessarily lead to high power output (hence, the effect of shear and veer may be interlinked). As a result, the impacts of the shear and veer on power output are

left for future work. Here, we simply highlighted the spatiotemporal variability of shear and veer and treated them as a source of uncertainty in wind resource modeling, that must be minimized. We feel confident in this assessment as the transition from high to low shear conditions across the lease areas from ocean to coast follows an increasing probability (~30 to ~40%) of finding at least the top half of the wind turbine rotor layers in the free troposphere (Supplementary Figure S8). Wind energy models typically

assume the turbine location within the well-mixed boundary layer, whereas free tropospheric winds are more influenced by larger-scale (mesoscale or synoptic) processes.

One limitation of this study stems from the use of reanalysis instead of observational data, which may not capture all small-scale meteorological features and can introduce uncertainties in the representation of wind speeds and boundary-layer processes. Additionally, while FV and AV provide valuable insight into

variability, they are influenced by the specific choice of wind turbine model (in this case, the IEA 15 MW RWT). Future studies may benefit from evaluating different turbine models and examining their sensitivity to the metrics used here. However, Supplementary Figures S17 and S18 does provide preliminary evidence that such a choice of wind turbine power curve might not alter the locations of optimal wind resource at either WEA significantly.

Another limitation of the study is the usage of hub-height wind speeds instead of a more complicated rotor-layer equivalent wind speed being adopted by the wind industry (which also happens to be more capable of accounting for the effects of veer and shear). Although our choice of hub-height wind was because of simplicity and the absence of observed vertically resolved wind profiles, we should note that we are likely not very far off in our estimates. For example, Van Sark et al. (2019) showed that when the

ratio of rotor diameter and hub height is smaller than 1.8 (ours is 1.6) and the wind shear coefficient varies between −0.05 and 0.4 (>60% of the distributions of Supplementary Figures S10), these two estimates of wind speeds only vary within 1%. However, a detailed analysis considering the impact of rotor-layer equivalent winds will also be the subject of our future analysis.





Moreover, the analysis did not account for the effects of wind turbine wakes, which can significantly impact power output variability in real-world wind farms. Finally, a detailed analysis of ramp events and wind extremes across the lease areas will be critical for understanding the potential for wind floods, droughts, and other extreme events that could impact power generation stability and potentially alter the maps of optimal power output that has been presented in this study. Hence, it is advisable to treat the linear optimization presented here as the first of a series of steps that need to be undertaken to determine the ideal layout of wind turbines in the offshore wind energy areas at Humboldt and Morro Bay. Quantifying the impact of turbine wakes on the power, and characterizing the occurrence and nature of extreme events in the two WEA will be focus on our future work.

**Supplementary Materials**

Additional figures are included in Supplementary Materials.

*Author Contributions.* **Conceptualization:** All authors; **Data curation:** All authors; **Formal analysis:** AM; **Funding acquisition:** VG; **Investigation:** AM and VG; **Methodology:** AM and VG; **Project administration:** RK and VG; **Resources:** AM and VG; **Software:** AM; **Supervision:** VG and AM; **Visualization:** AM; **Roles/Writing - original draft:** AM; **Writing - review & editing:** All Authors

*Data Availability.* The NOW-23 dataset was downloaded from https://data.openei.org/submissions/4500. The 15 MW Reference Wind Turbine (RWT) Power Curve was taken from: https://nrel.github.io/turbine-models/IEA_15MW_240_RWT.html.



*Competing interests.* The corresponding author has declared that none of the authors has any competing interests.

*Acknowledgements.* This research was funded by the Energy Efficiency Renewable Energy (EERE)'s
Wind Energy Technology Office (WETO) through a contract DE-AC02-06CH11357 awarded to Argonne National Laboratory. Pacific Northwest National Laboratory (PNNL) is operated by Battelle Memorial Institute for the U.S. Department of Energy under Contract DE-AC05-76RL01830. We gratefully acknowledge the computing resources provided on Bebop; a high-performance computing cluster operated by the Laboratory Computing Resource Center (LCRC) at the Argonne National Laboratory.
The authors gratefully acknowledge Thomas Surleta for downloading and curating much of the data used in this study.

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
