# Peer review of "Locating the Optimal Wind Resource within two Californian Offshore Wind Energy Areas"

_Wind Energy Science, 2025_

## Author Comment (AC3)

**Response to Review of WES-2025-55**

We thank the reviewers for their comments and the editor for the opportunity to address the comments and work on a revised manuscript. Two key concerns common to both reviewers are addressed jointly immediately below, whereas point-by-point responses to their individual comments are provided further below. The comments are in black, and our responses are in blue. Any changes to the article text are mentioned in Italics below each comment.

The common concerns raised by both reviewers and the response are as follows:

**a) Wake Effects**

As mentioned in the original submitted manuscript (e.g., Lines 70-71 and 387-390), we had initially made a calculated choice to only use "idealized power" for this particular study without incorporating the impact of wind turbine wakes. We had chosen this approach as we had believed that the true offerings of this study were: a) the Fractional variability (FV) metric for variability quantification and b) the linear spatial optimization method to determine the location of the optimal wind-resource within a pre-construction wind energy area. We had further felt that including a discussion on wind turbine wakes would overcomplicate the narrative flow of the paper and draw attention away from these key offerings. Hence, we had repeatedly addressed the "idealized power" consideration as a shortcoming of the current study that (as mentioned in the Conclusion) we would address in a follow-up paper. However behind-the-scenes, we had already determined the relative percentages of power output lost to wind turbine wakes using an engineering wake model (PyWake as suggested by Reviewer-1) even before this manuscript was submitted and their spatial patterns were, in fact, similar to what the reviewer had anticipated.

Given both the reviewers' suggestions, we have reconsidered our approach, and we have introduced a new Section 3.3 immediately following the discussion on "Spatiotemporal Variability of Daily Aggregate Energy". This section is now titled "Deviations from Idealized Energy Assumptions". This section now includes a discussion on the spatial variability of rotor-layer wind shear/veer (moved from Section 3.2) and wind turbine wakes. The previous Section 3.3 has now been moved to an Updated Section 3.4, which now includes an updated linear optimization score

which now also minimizes for wind turbine wake losses (apart from the other objectives). To accommodate the changes, further updates were made to the manuscript as listed below:

- The Abstract acknowledges wake-induced losses as an objective to be minimized in the spatial optimization.
- The Introduction now highlights that Wind turbine wakes are a consideration over and above the "idealized power" estimates in our investigation for the location of optimal wind resource.
- Section 2.1 accounts how the wind turbine effects are estimated for the Humboldt and Morro Bay WEAs for various weather regimes.
- Updates to Sections 3.3 and 3.4 are mentioned above.
- The Conclusion has also been updated to account for our new results in the light of wakeinduced losses.

**b) Supplementary Material**

Both reviewers had highlighted that they felt that certain key figures had been relegated to the Supplementary Material, whereas logically they belonged either in the main text or in the Appendix to the main text. We concur with their assessment and thank them for their suggestion. The following changes have now been implemented:

Appendix Figure A1 – Power Curve and Rate of Change of Power Curve with Wind Speed

Appendix Figure A2 – Seasonal Wind Roses for Humboldt and Morro Bay to aid the discussion from Section 3.1

Figure 10 - The figures highlighting the sensitivity test of the linear spatial optimization technique to randomized changes in wind turbine power curve choice is combined into one figure, now in the main text.

We still have a rather extensive Supplementary Material which depicts the seasonal variabilities of some of the annual averages shown in the main text (Energy, FV of energy, shear/veer, weather regimes, MIN/MAX output, etc.), as including such figures in the main text would complicate the narrative. However, we are grateful to the reviewers for their suggestions, as including the erstwhile Supplementary Figures in the Main text has improved the readability of the paper.

**Reviewer 1:**

The manuscript "Locating the Optimal Wind Resource within two Californian Offshore Wind Energy Areas" by Mitra et al. presents in general quite interesting and relevant research on offshore wind resources in California. The further investigation of potential future offshore lease areas is from a scientific and industrial perspective very relevant.

**Thank you for highlighting the relevance of the study.**

However, I found several key deficiencies in the manuscript that lead to the conclusion that I can't recommend publication of the manuscript in wind energy science. Those points are - from my perspective - so severe that a revision of those points would lead to a completely different manuscript and results.

We appreciate your feedback and want to share that we have managed to incorporate all of your key suggestions, without the necessity of "a completely different manuscript". Our goal was initially to develop a subsequent article focusing on additional analyses, particularly concerning wake effects. However, we demonstrate that integrating these new findings did not necessitate a resubmission of the current article. However, we acknowledge that incorporating these results would indeed enhance the current article, so we thank you and Reviewer-2 for your suggestions.

Specifically, I have identified the following shortcomings:

- The authors ignore wake effects in their design of the study. Thus, in principle, they are investigating the purely academic question of placing and moving around a single wind turbine in the two wind energy development areas. This results in a conclusion that the most optimal areas are for instance located at the southern edges of the wind farm, in areas with a very pronounced northerly wind component (see wind roses in supplementary material). In a real wind farm (and those are prospective real sites) those areas would be the ones with the strongest wake effects. Wake effects can lead to 10-30% yield loss, in extremely dense wind farm clusters even more. As the variability of the annual averaged daily energy output (Fig. 1) across the sites is in the same order of magnitude, the results would completely shift. There are layout optimization tools like floris, FOXES, pywake/TOPFARM that could be used for including wake effects in the

optimization. Or at least using a geometric wind farm layout and an engineering wake model would help to increase the relevance of the results.

This point had been addressed above as a Key Concern.

To summarize, the points you had raised were extremely relevant and your intuitions were onpoint. We did use the native 2-km grid to place a wind-turbine at each grid-cell-center and used PyWake (with the Bastankhah Gaussian engineering wake model) to estimate wake losses as functions of hub-height wind speed and directions. Indeed, we found that wakes result in about 15-30% losses in output energy, under various weather regimes. When this is considered in the linear spatial optimization technique, this indeed pushes the most optimal wind resource location not at the southern, but rather the northern flanks of the WEAs, as you had predicted.

- The authors are making some random choices on data processing, which are not explained. For example:

Why is there a resampling of the 2 km grid to a 0.02-degree reference grid?

Thank you for highlighting this issue. This was a remnant from an earlier iteration of the manuscript when we had included more data sources (other than NOW-23). To ensure proper sampling, we had interpolated all to the 0.02-grid (which roughly comes out to a 2-km spatial grid). However, while we had not excluded the other data sources from our final study and used the NOW-23 product at its native grid (without resampling), the earlier resampling had unfortunately been retained in the text. Now, all references to the resampling has been removed from the text.

Why are there 25 random turbine power curves investigated by varying cut-in, cut-out and rated power where also other reference turbines could be used such as DTU 10 MW turbine, the IEA 22MW turbine, the IEA 5MW turbine or even publicly available power curves for some existing turbines (a web search can help) could be used?

We thank the reviewer for this comment. We opted to generate 25 randomly varied turbine power curves (by perturbing cut-in, cut-out, and rated wind speeds and powers) to provide a flexible and generalizable sensitivity test. This approach allows us to evaluate the robustness of our

optimization framework not just for current standard reference turbines, but also for potential future turbine designs operating within a similar power class (8–16 MW). We note that the DTU 10 MW turbine, IEA 5 MW, and IEA 22 MW turbines – as well as other publicly available designs – fall within or near the range spanned by the random curves we used. Thus, our ensemble of power curves includes representatives that are functionally comparable to existing turbines. This randomized approach avoids overfitting to any single model and helps demonstrate that the spatial structure of optimal locations is largely insensitive to specific turbine choice, provided the rated power remains within this plausible offshore range.

- Why is the variability discussion not based on an annual energy production but on daily output variability? The first one would be an established measure from the wind energy community.

This was addressed in the introduction:

"Wind speeds and the electric power generated from wind farms can vary across different spatiotemporal scales. Hence, in recent years, a lot of emphasis has been put on the ready availability of short-term (sub-hourly to daily) wind speed forecasts at or near wind-turbine hub heights and wind turbine rotor layers (Wilczak et al., 2019; James et al., 2022). As a result, such forecasts are now readily available from direct numerical simulation and statistical postprocessing. Variability within these forecasts on hourly and daily timescales are largely driven by the inherent diurnal variability of the Earth's climate (occasionally disrupted by random weather events, such as surface low-pressure systems). Hence, characterizing this natural daily variability and the deviations from the same is crucial in designing the layout of wind turbines, estimating the electricity produced by wind farms over their lifetime and in devising strategies to improve the lifetime of the farms in question."

We did not seek to quantify the interannual variability but design a metric to quantify small-scale variability that would be relevant to a more holistic decision-making for future wind farm designs. This point has been further illustrated in the revamped introduction section of the updated manuscript.

- There is no discussion at all on available reference measurements in the area, or how they went into the NOW-23 dataset, if they are sufficiently included there.

There are several reference measurements in the area, including 1-year long DOE lidar buoy observations at Humboldt and Morro Bay wind energy areas (Krishnamurthy et al., 2023). The reference measurements from various offshore surface-based buoys (from National Data Buoy Center) and ground-based observations (including Radars, surface met stations etc.) are compiled here: https://wdh.energy.gov/project/oracle (a free database of wind observations). These observations were used to fine tune and evaluate the NOW-23 model outputs in existing literature (Liu et al., 2024, Bodini et al., 2024). However, a detailed discussion of the reference measurements is not included in the text itself as it not directly relevant to our study.

Krishnamurthy, R., García Medina, G., Gaudet, B., Gustafson Jr., W. I., Kassianov, E. I., Liu, J., Newsom, R. K., Sheridan, L. M., and Mahon, A. M.: Year-long buoy-based observations of the air–sea transition zone off the US west coast, Earth Syst. Sci. Data, 15, 5667–5699, https://doi.org/10.5194/essd-15-5667-2023, 2023.

Liu, Y., Gaudet, B., Krishnamurthy, R., Tai, S. L., Berg, L. K., Bodini, N., ... & Kumler, A. (2024). Identifying meteorological drivers for errors in modeled winds along the Northern California Coast. Monthly Weather Review, 152(2), 455-469.

Bodini, N., Optis, M., Redfern, S., Rosencrans, D., Rybchuk, A., Lundquist, J.K., Pronk, V., Castagneri, S., Purkayastha, A., Draxl, C. and Krishnamurthy, R., 2023. The 2023 National Offshore Wind data set (NOW-23). Earth System Science Data Discussions, 2023, pp.1-57.

- The literature review refers at several points to a paper Mitra et al., 2024, which is not part of the list of references. There, an unpublished study (Mitra et al., 2025) is mentioned, which I couldn't find even in a web search. This also this part would need some very thorough revision.

We had hoped to have that manuscript submitted to Wind Energy Science, however, it was found to not fully adhere to WES scope. So, it is in the process of being released publicly as an Argonne Technical Report with an US DOE-OSTI DOI number. However, there has been a recent backlog in the US DOE OSTI Technical Report releases. While we have been provided the DOI number at which the report will be released, we do not have a clear communication on when it will be made publicly available. However, it has been currently authorized for release by the Argonne National Laboratory. The updated citation highlights the current status of that manuscript: "Mitra, A., Ghate, V., Krishnamurthy, R.: Wind and Climate Variability within the Californian Offshore Wind Energy Areas, Argonne Technical Report, ANL/NSE-24/26, Pending Release at U.S. Department of Energy (DOE) OSTI, https://doi.org/10.2172/2568064, 2025 [Currently accessible online at: https://publications.anl.gov/anlpubs/2025/05/196524.pdf]."

- The manuscript has quite a long document of supplementary material, which is at some points very relevant for the interpretation of the results, such as the used power curve, or the wind roses. The choice of what went into supplementary material seems quite random to me. I would recommend putting them rather into an appendix or at some points even in the main part of the document.

Thank you for this suggestion and this has been addressed as a Key Concern earlier. According to your suggestions, we have moved a couple of figures from the Supplementary Materials to the Appendix.

**Reviewer 2:**

This manuscript addresses an important and timely topic in offshore wind energy by examining the spatiotemporal variability of wind resources at two proposed Californian lease areas using long-term, high-resolution reanalysis data. The effort to integrate wind variability, shear, and veer into a spatial optimization framework is commendable, and the analysis benefits from a rich dataset and a relevant application to real-world siting challenges.

**Thank you for highlighting the relevance of the study.**

However, despite these strengths, the manuscript in its current form is not suitable for publication. The methodological foundation is underdeveloped, key concepts are either misapplied or oversimplified, and the paper lacks the necessary context and rigor expected for a scientific contribution in this area. Major revisions would not be sufficient to address the fundamental flaws. Rather, a substantial reworking of the manuscript, both in terms of technical approach and scientific framing, is required. What is needed is effectively a new paper: one that clearly engages with the existing literature, justifies methodological choices, incorporates essential physical processes such as wake interactions, and includes a proper discussion of limitations and implications.

We appreciate your feedback and want to share that we have managed to incorporate all of your key suggestions, without the necessity of "effectively a new paper". Our goal was initially to develop a subsequent article focusing on additional analyses, particularly concerning wake effects. However, we demonstrate that integrating these new findings did not necessitate a resubmission of the current article, rather the inclusion of analyses which we had already done and rewording and clarifying some earlier confusing sections. However, we acknowledge that incorporating these results has enhanced our manuscript, so we thank you and Reviewer-1 for your suggestions.

**General remarks**

The manuscript relies heavily on supplementary material to present several key results, including figures that are essential to understanding and evaluating the methodology and its outcomes. While supplementary materials can be appropriate for secondary or supporting analyses, critical figures, such as those demonstrating the sensitivity of optimization results to turbine characteristics or illustrating variability patterns, should be included directly in the main manuscript.

Thank you for this suggestion and this has been addressed as a Key Concern earlier. According to your suggestions, we have moved a couple of figures from the Supplementary Materials highlighting the sensitivity analysis of the optimization and combined it as Figure 10 of the Main Text.

**Methodology**

The methodology presented in the preprint suffers from several critical limitations that undermine the robustness and practical applicability of the results. First and foremost, the choice of weights in the multi-objective optimization function is arbitrary and lacks any empirical or stakeholderdriven justification. All weights are set to one (with a slightly higher value of 1.5 for distance to shore), implying equal importance of energy output, variability, shear, veer, and cost, which is rarely the case in real-world wind farm planning. Because the optimization score is highly sensitive to these weights, the interpretation of optimal locations is equally sensitive and thus not reliable without a thorough sensitivity analysis or justification of trade-offs.

The choice of initial weights is indeed arbitrary, as with any optimization models. But these are only the initial weights and not their final values. However, the weights are not fixed but rather updated during the course of the gradient descent operation to find the minima of the cost function. As a result, the operation is indeed in a classical sense a "gradient descent optimization". These points will be made clearer in the resubmission, through clarifications in the updated Section "Location of Optimal Wind Resource" (now Section 3.4).

Compounding this issue is the misuse of terminology: the authors refer to their method as "gradient descent optimization," but no optimization is performed in the classical sense. The weights are fixed, and the scoring is applied statically to each grid cell, more akin to a composite ranking or multi-criteria evaluation than an iterative optimization process.

The algorithm detailed in this study is indeed a gradient descent-based optimization in the classical sense and the weights are not frozen. To avoid any confusion, we have also clarified that the algorithm we have implemented is a "multi-objective scoring framework optimized via gradient descent" at in the first paragraph of Section 3.4. The iterative optimization component of the algorithm is now explained in more detail in the attached text (from Section 3.4):

"The algorithm iteratively optimizes the weights over 1000 iterations with a learning rate of 0.01. The objective is to maximize the total energy output over all grid cells, or  $\sum_i Out_i * (100 - wakes_i)\%$ , where,  $Out_i$  and  $wakes_i$  are the mean daily idealized energy output and wake-induced loss at each grid cell, respectively. To guide the optimization, we define the loss function as

$$L = -\sum_{i} S_i$$

where  $S_i$  is the optimization score function for a given grid cell, i. The gradients of this loss function are computed with respect to each weight (e.g.,  $\frac{\partial L}{\partial w_{shear}}$ ) and are used to update the weights via standard gradient descent. This procedure allows the optimizer to converge toward a set of weights that maximizes total suitability scores, which in turn corresponds to maximizing net energy production while minimizing penalties. The final raw scores were normalized to a 0-100 scale, with higher scores indicating grid cells most favorable for turbine placement." A far more serious flaw is the complete omission of wake effects, which are among the most important considerations in wind farm layout optimization. By treating each grid cell in isolation and assuming one turbine per cell, the methodology neglects the performance losses caused by turbine-turbine interactions, which can dramatically alter the spatial efficiency of a layout. As a result, the identified "optimal" locations may not remain optimal when array-level effects are considered.

This point had been addressed above as a Key Concern.

We have now used the native 2-km grid to place a wind-turbine at each grid-cell-center and used PyWake (with the Bastankhah Gaussian engineering wake model) to estimate wake losses as functions of hub-height wind speed and directions. We found that wakes result in about 15-30% losses in output energy, under various weather regimes. When this is considered in the linear spatial optimization technique, this indeed pushes the most optimal wind resource location not at the southern, but rather the northern flanks of the WEAs. These changes have been incorporated into the discussions in Sections 3.3 and 3.4.

Finally, the normalization of all variables to a 0–1 range without accounting for their relative physical or economic significance can distort the true influence of each factor on the final score. Taken together, these issues suggest that while the intent of the optimization framework is commendable, its execution lacks the methodological rigor and realism required for meaningful conclusions in the context of offshore wind development.

We respectfully disagree with the assertion that our normalization approach "distorts the true influence" of each factor. In multi-criteria optimization frameworks – particularly those operating on metrics with disparate units and ranges (e.g., MWh, %, degrees, km) – normalizing to a common scale (e.g., 0–1 or 0–100) is a standard and necessary step to ensure comparability and avoid numerical domination by any single term. For interpretability, we normalize each metric using min–max normalization across the WEA and apply explicit weights to reflect their relative influence on the final score. These weights are not fixed but are optimized iteratively using gradient descent, as now further clarified in Section 3.4. In effect, the algorithm *learns* the relative contribution of each term that best aligns with our objective function (net power output adjusted for wake losses). This guards against the arbitrary influence of scale alone.

We acknowledge that in the absence of stakeholder-defined priorities (e.g., explicit cost functions or regulatory constraints), the economic or physical "significance" of each metric cannot be fully codified. However, our goal here is not to produce a finalized wind farm layout or cost model, but rather to present a transparent, meteorology-informed ranking of turbine siting suitability under idealized conditions. Which is a novel and much-needed endeavor. The weights and scoring framework can easily be modified or extended in future work as more real-world constraints or priorities become available. We have clarified this intent in the revised text of Section 3.4 and in the final paragraph of the Discussion section.

**Introduction and discussion**

Another significant limitation of the manuscript lies in the introduction and discussion sections, which fail to adequately engage with the existing body of literature on methodologies for locating optimal wind resource areas. The introduction does not provide a review of current best practices or established benchmarks in wind farm sitting, such as multi-objective layout optimization frameworks that incorporate wake losses, cost models, or environmental constraints. This omission makes it difficult for readers to understand how the proposed method compares to state-of-the-art approaches in the field. A proper scientific contribution must situate itself within the existing landscape of methodologies, identify what gaps it intends to fill, and justify why its approach is advantageous or innovative. Moreover, no comparison is made between their simplistic scoring framework and more sophisticated optimization methods commonly used in industry and academia—such as genetic algorithms, greedy heuristics with wake modeling, or levelized cost of energy (LCOE)-based evaluations. The manuscript lacks any discussion of the limitations of their approach in relation to these established methods. The absence of such comparative analysis severely weakens the manuscript's credibility and relevance. A meaningful discussion section should critically reflect on the strengths and shortcomings of the proposed methodology considering existing literature, but this is entirely missing. As a result, the current discussion remains superficial and does not offer the reader a clear sense of the method's practical utility or scientific value.

We acknowledge the reviewer's concern regarding the lack of contextualization with existing literature in the originally submitted version. In response, we have substantially revised the

Introduction and Discussion sections to better situate our framework within the current landscape of wind farm siting methodologies. Specifically:

- We now contrast our method with widely used tools such as TOPFARM and FLORIS, emphasizing that while these frameworks are effective for *refining* turbine layouts under fixed inflow and boundary assumptions, they are not typically designed to assess *relative suitability* across the spatial heterogeneity of an undeveloped lease area.
- We also discuss cost-based screening approaches (e.g., LCOE-based evaluations) and explain how our method differs by incorporating not just long-term means, but also daily-scale variability and rotor-layer inflow metrics (shear, veer).
- In addition, we compare our approach to recent machine learning-based layout optimization studies, noting that while promising, such approaches often require high-fidelity training data not currently available for the California WEAs.
- Finally, we explicitly frame our method as a first-order, meteorology-informed ranking tool intended to bridge the gap between broad climatological screening and detailed CFD/micro-siting design stages (Introduction, Section 3.4 and Discussion).

We agree that this positioning was previously unclear and have addressed it through extensive changes throughout the manuscript. A fully revised framing now appears at the end of the Introduction (see final three paragraphs of Section 1), which now reads:

"This approach differs from conventional layout optimization tools (such as TOPFARM or FLORIS) which are typically used to optimize turbine positions under fixed site boundaries and inflow conditions (Larsen and Réthoré, 2013; Doekemeijer et al., 2021). While effective for refining turbine layouts once a wind farm design and turbine count have been established, such tools do not evaluate the relative quality of each turbine location across a lease area using multimetric, climatologically informed criteria. Moreover, our approach also departs from cost-based screening tools like Levelized Cost of Energy (LCOE), which rely on long-term average inputs and do not account for variability or rotor-layer dynamics. Other approaches using machine learning can also be effective in layout optimization (Yang and Deng, 2023; Yang et al., 2024) but need expensive high-resolution simulations as training datasets. Understanding the dynamic performance of floating offshore wind turbines in a coupled wind-wave-ocean-turbine sense is

also a topic of ongoing research (Froese et al., 2022; Mahfouz et al., 2024; Barnabei et al., 2024). However, we aim to fill the intermediate gap between broad-scale resource characterization and detailed micro-siting, by using long-term wind climatology to flag spatial heterogeneity within lease-scale WEAs. We posit that our approach is especially suited to locations such as the California offshore WEAs, where direct multiple years of rotor-layer or hub-height observations and real wind turbine data are absent, and the entire analysis and planning hinges on numerical model data. The guiding question is: "Given only a high spatial-resolution hourly wind profiles and a generic turbine design, what parts of a WEA offer the best balance of energy production and feasibility for initial turbine placement?" This framework provides a first-order decision-support tool for developers and stakeholders to prioritize turbine placement during the early stages of offshore wind farm design and permitting."

In the Discussion, we also reflect on the limitations of our approach relative to these established methods and clarify that our method is *not* intended to replace full-system layout optimizers or financial models, but rather to inform their early-stage application.

We thank the reviewer for this important observation and believe the revised manuscript now more clearly articulates both the niche and the utility of our proposed framework.